# Sociodemographic influences on children's out-of-school time organized activity participation and physical activity in rural communities: A cross-sectional study

Mary J. Von Seggern[1]*, Michaela A. Schenkelberg[2], Ann E. Rogers[1],
Debra K. Kellstedt[3], Robin High[4], Gregory J. Welk[5], Richard R. Rosenkranz[6],
David A. Dzewaltowski[1]

1 Department of Health Promotion, College of Public Health, University of Nebraska Medical Center, Omaha, Nebraska, United States of America, 2 School of Health and Kinesiology, College of Education, Health, and Human Sciences, University of Nebraska at Omaha, Omaha, Nebraska, United States of America, 3 Family and Community Health, AgriLife Extension, Texas A&M University, College Station, Texas, United States of America, 4 Department of Biostatistics, College of Public Health, University of Nebraska Medical Center, Omaha, Nebraska, United States of America, 5 Department of Kinesiology, College of Human Sciences, Iowa State University, Ames, Iowa, United States of America, 6 Department of Kinesiology and Nutrition Sciences, School of Integrated Health Sciences, University of Nevada, Las Vegas, Las Vegas, Nevada, United States of America

* mary.vonseggern@unmc.edu

## Abstract

Out-of-school time (OST) organized group youth activities (e.g., afterschool programs, clubs) can reduce health inequalities by increasing physical activity (PA). However, unlike youth sport, PA is not the primary focus of many organized activities, and little is known about rural children's non-sport organized activity participation. This study examined sociodemographic characteristics associated with OST non-sport organized activity participation and PA among children living in rural U.S. Midwest communities. During Spring 2019, children ($n = 418$ 3rd–6th graders) attending school in two rural communities completed a PA surveillance instrument as part of Wellscapes, a community randomized trial. Caregivers of a subsample of children ($n = 235$) consented to pair their child's survey results with student enrollment records. Mixed models with community as a random effect examined main and interaction effects of grade, sex, and family income on OST non-sport organized activity participation and these sociodemographic characteristics and organized activity participation on OST moderate-to-vigorous PA (MVPA) per weekday and weekend day. Most children (73.2%) participated in an OST non-sport organized activity. Males were less likely to participate than females (OR = 0.38, 95% CI = 0.20–0.73, $p = 0.004$). Females and 6th graders reported lower OST MVPA on weekdays and weekends than comparison groups ($p < 0.001$). Males with lower family income accumulated significantly fewer minutes of MVPA on weekdays ($mean_{diff} = -4.7 \pm 2.0$ minutes) and weekends ($mean_{diff} = -8.9 \pm 3.8$ minutes) compared with males with higher family

**Data availability statement:** Data for this and the larger Whole-of-Community Youth Physical Activity (Wellscapes) study are ethically restricted to protect child confidentiality. The Institutional Review Board (irbora@unmc.edu) and Sponsored Programs Administration (spadmin@unmc.edu) at University of Nebraska Medical Center forbid making the data publicly available because informed consent from study participants did not cover public deposition of data. However, the minimal data set underlying this study's findings is stored at the University of Nebraska Medical Center on a secure server and can be accessed by interested researchers on-site pending review of data usage application. Therefore, researchers who meet the criteria for access to confidential data may send their request to the University of Nebraska Medical Center, where the data is housed (coph@unmc.edu), or submit a request to Wellscapes Primary Investigator and senior author, Dr. David A. Dzewaltowski (david.dzewaltowski@unmc.edu).

**Funding:** This work was supported by the National Cancer Institute of the National Institutes of Health under Award Number R01CA215420. Study sponsors were not involved with data collection, analysis, interpretation, or writing of the manuscript. The content is solely the responsibility of the authors and does not necessarily represent the official views of the National Institutes of Health.

**Competing interests:** The authors have declared that no competing interests exist.

income ($p < 0.05$). Many rural children participated in OST organized activities regardless of grade and family income. However, there were disparities in organized activity participation and OST PA outcomes based on sociodemographic factors, including grade, sex, and family income. Designing OST organized activity settings to be more accessible and include opportunities for PA may help ensure children can achieve optimal health.

## Introduction

Physical activity (PA) is an essential modifiable health behavior that has emerged as a global public health priority to improve population health and alleviate healthcare system stressors [1]. Regular PA is linked to significant health benefits in children and adolescents, including improved mental health and academic performance and reduced risk of obesity, cancer, and other chronic diseases [2,3]. Additionally, PA behaviors during the school-aged years influence PA in adulthood, in turn, affecting the health of the general population [4,5]. Despite these connections, children and youth in the United States (U.S.) are insufficiently active [6]. According to recent surveillance estimates, less than a quarter of U.S. children aged 6–17 years participate in the recommended 60 minutes or more of moderate-to-vigorous physical activity (MVPA) every day [7,8].

Differences in PA levels among children defined by individual sociodemographic characteristics such as age, sex, socioeconomic status, and racial and ethnic minority status also persist [9–12]. Further, empirical studies have identified PA differences in community characteristics (e.g., rural and urban classifications) related to where children live, learn, and play [13–16]. Though evidence indicates adults living in rural areas are consistently less active than their urban and suburban counterparts, studies exploring community geographic-related PA differences among children report inconsistent results [17,18]. For example, Moore et al. [19] found differences in mean MVPA between rural and urban children ($n = 284$) attending middle school, with significantly lower levels of activity in rural children compared with urban children. Yet, other studies have found that rural children are more active than urban children [15–17,20,21]. These studies examined the differences in community residence and children's PA behaviors constrained by the classic rural-urban dichotomy [22] and did not account for PA variations within each community system, comprised of unique behavior settings children frequent.

Building upon Barker's definition of communities as systems of behavior settings, in which ecological units bounded by time and space drive patterns of behavior within a replicated social structure [23], the concern here is to forgo the classic rural-urban comparison and examine the population PA outcomes of children residing within rural communities. This approach emphasizes the need to move beyond the oversimplified rural typology and embrace the complexity of rural communities and the system of organized settings (e.g., school classrooms, youth sport practices, and club meetings) nested within, particularly given the variability of available and accessible

settings across rural communities [24,25]. Population PA outcomes for children are highly dependent on the interactions among individuals within setting environments (e.g., peer influence and leader behavior) and among the community "wellness landscape" [26] of organized settings children frequent, including school and out-of-school time (OST) organized settings, with different structuring properties (i.e., rules and resources) [27–29]. Bronfenbrenner's ecological theory [30] further aligns with this approach and the need to study the complex interplay between children, their immediate environments, and the larger whole-of-community system [31]; however, much of the literature has examined children's PA behaviors solely within the school setting, often isolating individual outcomes and neglecting the places children go during OST [32,33]. Thus, investigating the places rural children frequent outside of the school day is critical to understanding their PA behaviors.

Evidence also indicates some children are spending more time participating in the heterogeneous landscape of community OST organized settings that feature a range of adult-led activities, which may help to explain the current decline in child-driven free play [34]. These organized settings often include afterschool programs and extracurricular activities (e.g., clubs and youth sports), which typically meet both before and after school, in the evening, and on weekends [35–38]. Provided PA outside of the school day makes a substantial contribution (i.e., almost half) to a child's daily total PA [39] and children are spending more time in OST school and community-based opportunities, these settings hold promise for influencing childhood behaviors, including widespread PA promotion efforts [40]. However, OST organized activities vary widely within local communities and across the U.S., serve many different purposes (e.g., academic, youth development, enrichment, or childcare), and rely on different funding mechanisms (e.g., public or private sources) to cover programming costs, not limited to staff time or necessary equipment [41].

Organized youth sport, where PA is a primary purpose, has been identified as an effective public health strategy to increase childhood PA [42], but the current pay-to-play model with an estimated average $883 annual price tag per child and sport [43], among other factors (e.g., overemphasis on sport specialization and winning-at-all-costs), has erected barriers to participation [44,45]. For example, the presence of youth sport participation barriers among children living in rural Great Plains communities was recently highlighted by Kellstedt et al. [46] and Von Seggern et al. [47], in which children from higher-income families were almost four times more likely to participate in youth sport than their lower-income peers, and non-Hispanic White children were over five times more likely to participate than Hispanic children, respectively. Thus, understanding if non-sport OST organized opportunities are accessible, regardless of primary purpose, is fundamental as these often-overlooked opportunities with substantial cost variability (e.g., within Scouting, cost estimates range anywhere from $110 to over $600 annually per child) [48] may be critical to reaching more children for PA-promotion efforts, including those priced out of youth sports.

Despite the potential reach of these settings, the contribution of non-sport organized activity involvement, such as participating in clubs and youth organizations (e.g., 4-H, Scouting, and STEM), to children's PA is not well understood [36,49]. Further, even less is known about non-sport OST organized activity participation and PA among rural children, although patterns of involvement in these activities are beginning to emerge in the literature [25,50,51]. Evidence suggests settings where PA is not the primary purpose (e.g., schools and afterschool programs) can improve children's PA by inserting opportunities for PA (e.g., physical education classes, outdoor play, and brain breaks) or adopting PA policies, like the National AfterSchool Association's [52] healthy eating and physical activity (HEPA) standards or other active recreation policies [53–55]. However, Sliwa et al. [35] acknowledged that many OST organized opportunities lack PA policies and practices or awareness of existing policies, particularly in those administered outside of schools and school districts. Therefore, understanding the places children go outside of the school day, the quantity of the PA provided by these opportunities, and how these and other factors (e.g., sociodemographic characteristics) interact to influence childhood PA outcomes, will help to inform non-sport setting design and better support the integration of public health policies, systems, and environmental change strategies related to PA-promotion efforts, explicitly in understudied and underserved rural communities. As a result, the goals of this study were to 1) examine the influence of grade, sex, and family income

on participation in community non-sport OST organized settings, and 2) determine the influence of these factors (i.e., grade, sex, family income, and non-sport OST organized activity participation) on the OST PA levels of children living in rural communities.

## Methods

### Study design

This study was a cross-sectional sub-study of the Whole-of-Community Youth Physical Activity (Wellscapes) Project. The Wellscapes Project was a two-wave staggered-start community randomized trial (ClinicalTrials.gov Identifier: NCT03380143, 20/12/2017) and social epidemiology study of four rural U.S. Midwestern communities aimed at establishing a whole-of-community multilevel system infrastructure to increase youth population PA [24]. The protocol for all study activities was approved by the Institutional Review Board (IRB) at University of Nebraska Medical Center (IRB #446–18-EP, IRB #439–18-EX).

In Wave 1 (fall 2018-spring 2020), two rural, predominantly non-Hispanic White communities were targeted for planned recruitment based on population size (i.e., less than 3,500 residents) and proximity to urban areas (i.e., distance greater than 10 miles from an urbanized area) [56]. In Wave 2 (fall 2021-spring 2023), two rural communities with a majority of Hispanic children (i.e., school districts with a [≥50%] concentration of Hispanic children) were recruited. Additional inclusion criteria for Wave 1 and Wave 2 communities included the following: had one public high school, offered multiple adult-led group opportunities (e.g., youth club meetings, youth sport practices, etc.) for children in 3rd through 6th grades (approximately 8–12 years of age), independent completion of a community health needs assessment with prioritization of obesity prevention in a community health improvement plan, and agreement by the local health department and public school district to participate in the study.

### Study sample

The present study reports on spring 2019 population PA and corresponding contextual data collected during the Wellscapes Wave 1 baseline year. With permission from the school districts ($n=2$), administration, and staff, all 3rd through 6th-grade public education classrooms ($n=21$) across communities ($n=2$) were eligible to participate in Wave 1's social epidemiology study, which consisted of the administration of an online PA surveillance instrument near the beginning and end of each school year during designated class time. The instrument is comprised of the validated Youth Activity Profile (YAP) [57,58], an online assessment explicitly designed for school-based evaluations of youth PA with 15 items that children and adolescents can complete in 15–20 minutes [59]. Students in 4th grade and up can complete the YAP independently [57,59]; however, children as young as 3rd grade can complete the YAP with adult supervision as documented in the National Cancer Institute's Family Life, Activity, Sun, Health, and Eating (FLASHE) Study [60]. The instrument also includes supplemental out-of-school organized activity participation questions from the National Survey of Children's Health 2017-2018 (NSCH) for comparison of results at multiple levels (i.e., national, state, and local levels) [61]. We specifically concentrated on the second administration of the instrument in May 2019, as opposed to the first administration in September 2018, to capture children's self-reported PA behaviors that occurred during the school year. Previous publications have documented the protocol [62] and summarized descriptive PA and youth sport participation patterns at the population level in the Wave 1 communities [46].

A total of 465 3rd through 6th-grade children across both communities were eligible for participation in the social epidemiology study as part of normal educational practice and deemed exempt by the IRB at University of Nebraska Medical Center (IRB #439–18-EX), of which 418 provided assent and completed the PA surveillance instrument (90%). This study primarily focused on a subset of those children ($n=235$) who had informed parental and guardian written consent to participate in the Wellscapes community randomized trial and pair children's school sociodemographic data (e.g.,

date of birth, sex, free and reduced lunch status, race, and ethnicity) with their respective PA data, including surveillance instrument responses, based on a Data Sharing Agreement (DSA) established with participating schools [62]. Members of the research team made multiple attempts (e.g., working with teachers and organized activity leaders), beginning September 05, 2018, and ending May 31, 2019, to gather informed parental and guardian written consent for community trial participation. The IRB at University of Nebraska Medical Center approved data collection procedures in 2018 (IRB #446–18-EP).

## Procedures

The research team instructed school administrators and teachers to administer the online instrument as an in-class educational experience (e.g., to learn about PA and sedentary behaviors) and provided ongoing technical assistance. Coinciding with the conclusion of the baseline year, the surveillance instrument was administered during the first week of May 2019 in both communities. Children were guided through the online instrument with support from teachers and school staff but self-reported grade, sex, organized activity participation, and PA and sedentary behaviors. All 3rd through 6th graders completed the YAP and supplemental questions within a classroom setting using media carts or the school's media center. Children were instructed to complete the instrument using individualized login information [62].

## Measures

**Sociodemographic characteristics.** Individual sociodemographic characteristics in this study included grade, sex, and a proxy for family income (i.e., lunch status) based on student eligibility requirements for free or low-cost meals during the school day (e.g., children from families with incomes at or below 130% or between 130% and 185% of the federal poverty line, respectively) as part of the National School Lunch Program [63], a U.S. Department of Agriculture federally assisted meal program operating in public and private schools and residential child care institutions across the nation [64]. When completing the YAP, children were asked to self-report grade (i.e., 3rd, 4th, 5th, or 6th grade) and sex (i.e., male or female). The DSA permitted the school to provide additional sociodemographic information obtained from school enrollment records, including student name, identification number, date of birth, grade, sex, race (i.e., White, Black/African American, American Indian/Alaska Native, Asian, Multiple Races, and Unknown/Did Not Report), ethnicity (i.e., Hispanic and non-Hispanic), and lunch status (i.e., free, free direct certification, reduced, reduced direct certification, and full pay or reimbursable). The school shared these data on a secure platform with the research team, and members of the research team paired de-identified school enrollment records data with surveillance data for the subsample of children whose parents consented to the community trial [62]. Given our target population was predominantly non-Hispanic White, race and ethnicity variables were combined (i.e., race/ethnicity) and dichotomized into two categories: non-Hispanic White and all other ethnoracial classifications (i.e., Hispanic White, and non-Hispanic and Hispanic: Black/African American, American Indian/Alaska Native, Asian, Multiple Races, and Unknown/Did Not Report). The family income variable was also dichotomized based on family free and reduced-price lunch status (FRLS) and full pay lunch status [65]. Consistent with our previous work, children with FRLS were considered "lower income," and children with full pay lunch status were considered "higher income" [46]. Within the constraints of the student enrollment records and DSA, we acknowledge lunch status is a sufficient proxy for family income based on the available data [66].

**OST organized activity participation.** OST organized activity participation was determined using the supplemental NSCH questions added to the surveillance instrument [61]. Specifically, these items assessed participation in non-sport out-of-school activities, including daily afterschool programs, clubs or organizations, and other organized activities or lessons. Children reported "yes" or "no" responses to the following questions: (1) "During the past 12 months, did you participate in a daily afterschool program?", (2) "During the past 12 months, did you participate in any clubs or organizations (4-H, Scouting) after school or on weekends?", and (3) "During the past 12 months, did you participate

in any other organized activities or lessons, such as music, dance, language, or other arts?" [61]. A new variable was then created to determine participation in any OST non-sport organized activity (e.g., afterschool programs, clubs, organizations, art lessons, etc.). Specifically, if a child answered "yes" to any of the three previously mentioned questions, they were included as an OST (non-sport) organized activity participant.

**Moderate-to-vigorous physical activity.** The primary outcome variables were determined using the YAP and included average weekday (Monday–Friday) minutes per day of OST MVPA and average weekend minutes per day of MVPA. These variables were obtained using published calibration equations developed specifically for the online version of the YAP [57].

In the YAP, children respond to five questions about in-school PA, five questions related to out-of-school PA, and five questions pertaining to sedentary behaviors. Average PA accumulated per weekday and weekend days is then estimated using an algorithm converting raw YAP scores into minutes of PA per day [57]. For the purpose of this study, the YAP items used to estimate weekday OST MVPA asked children to report how many days (0, 1, 2, 3, or 4–5 days) they did some form of PA for at least 10 minutes 1) before school (between 6:00–8:00 a.m.), 2) after school (3:00–6:00 p.m.), and 3) in the evening (6:00–10:00 p.m.), Monday – Friday. Weekend MVPA was determined by the YAP items that asked about amounts of PA: 1) no activity (0 minutes), 2) small amount of activity (1–30 minutes), 3) small-to-moderate amount of activity (31–60 minutes), 4) moderate-to-large amount of activity (1–2 hours), and 5) large amount of activity (more than 2 hours), accumulated on Saturday and Sunday (e.g., exercise, work/chores, and family outings). These raw YAP items were used in the calibrated regression-based algorithms [57] to estimate group weekday OST MVPA and weekend MVPA, respectively. The YAP calibration methodology [67], according to Welk et al. [57], has a unique advantage in that "the generated equations predict the percentage of time spent in PA for each item," allowing for MVPA estimates for distinct portions of the day and the accommodation of different segment durations. Specifically, daily weekday OST estimates were averaged to provide a group-level estimate (e.g., estimate for school-aged children by grade) for weekday OST MVPA, and daily weekend estimates were averaged to provide a group-level estimate for weekend MVPA.

When paired with PA measured by a validated PA monitor (i.e., a SenseWear Armband Pro3), the YAP has shown a low to moderate (r = 0.19–0.58) level of correlation [67]. Further, group-level estimates of in-school and out-of-school PA values from the YAP were within 23% and 21% of values derived from the PA monitor, respectively, based on mean absolute percentage error calculations [57].

## Data analyses

Descriptive statistics were used to summarize children's participation in organized activities for the social epidemiology study (n = 418) and the subset of children included in the community trial (n = 235). Mixed-models were used to analyze the dichotomous outcomes (i.e., 1 = Yes; 0 = No) of any OST organized activity, afterschool program, club, and other organized activity (e.g., music and dance) participation, and the continuous outcomes of average daily minutes of OST MVPA on the weekdays and average daily minutes of MVPA on weekend days for community trial participants only. For all models, community was included as a random effect, and race/ethnicity was excluded as a variable of interest (i.e., group counts were too low for all other ethnoracial classifications in comparison to predominantly non-Hispanic White study sample). SAS Studio and SAS/STAT software, version 9.4 (© 2002–2012) of the SAS System for Windows (Cary, NC), were used for all analyses.

PROC GLIMMIX was performed to examine the dichotomous outcome of any OST non-sport organized activity participation with grade, sex, and family income and their interactions used as fixed effects. PROC MIXED was used to examine weekday OST and weekend MVPA with grade, sex, family income, and any OST organized activity participation (as well as afterschool program, club, and other organized activity participation) and their interactions used as fixed effects. Backward elimination of non-significant interactions based on statistical significance of p < 0.05 was used where non-significant higher order interactions were eliminated first, and then the models were refit [68,69].

## Results

Descriptive characteristics of children in the Wellscapes social epidemiology study and those who participated in the Wellscapes community trial are found in Table 1. A total of 418 3rd through 6th-grade children completed the YAP, and of those, 235 children had parental consent to be included in the community trial. Among the community trial participants, 73.2% participated in at least one OST organized activity, and 75.3% were considered to have a higher family income based on full pay lunch status.

## OST organized activity participation

In both the social epidemiology and community trial studies, more children participated in at least one OST organized activity than did not (67.9% and 73.2%, respectively). Among community trial participants, 37.4%, 43.4%, and 47.2% of

**Table 1. Descriptive characteristics of participants.**

|  | Social Epidemiology Study Participation | Community Trial Participation |
|---|---|---|
| Grade in School | n (%) | n (%) |
| 3rd – 6th | 418 | 235 |
| 3rd | 108 (25.8) | 56 (23.8) |
| 4th | 100 (23.9) | 56 (23.8) |
| 5th | 117 (28.0) | 61 (26.0) |
| 6th | 93 (22.2) | 62 (26.4) |
| Sex |  |  |
| Male | 209 (50.0) | 117 (49.8) |
| Female | 209 (50.0) | 118 (50.2) |
| Race/Ethnicity |  |  |
| Non-Hispanic White | -- | 220 (93.6) |
| Hispanic/Racially Diverse* | -- | 15 (6.4) |
| Family Income |  |  |
| Lower (Free/Reduced) | -- | 58 (24.7) |
| Higher (Full Pay) | -- | 177 (75.3) |
| Daily After School Program** |  |  |
| Yes | 142 (34.0) | 88 (37.4) |
| No | 276 (66.0) | 147 (62.6) |
| Club or Organization** |  |  |
| Yes | 159 (38.0) | 102 (43.4) |
| No | 259 (62.0) | 133 (56.6) |
| Other Organized Activity** |  |  |
| Yes | 183 (43.8) | 111 (47.2) |
| No | 235 (56.2) | 124 (52.8) |
| Any OST Non-Sport Organized Activity** |  |  |
| Yes | 284 (67.9) | 172 (73.2) |
| No | 134 (32.1) | 63 (26.8) |

*Includes the following: Hispanic White, and non-Hispanic and Hispanic: Black/African American, American Indian/Alaska Native, Asian, Multiple Races, and Unknown/Did Not Report

**Participation in past 12 months

children specifically reported participating in daily afterschool programs, clubs, and other organized activities (e.g., music lessons), respectively.

Table 2 shows the results from the mixed-model regression predicting OST organized activity participation among community trial participants. Compared to females, males were significantly less likely to participate in any OST organized activity (OR = 0.38, 95% CI = 0.20–0.73, $p = 0.004$). There were no significant differences in the likelihood of participating in any OST organized activity by grade level or family income.

The differences between OST organized activity participation and average daily minutes of OST MVPA per weekday and weekend among community trial participants are highlighted in Fig 1. Children who participated in daily afterschool programs reported significantly ($p = 0.014$) more daily minutes of OST MVPA per weekday (mean = 56.1 ± 1.1 min/day) than

**Table 2. Likelihood of participating in any OST non-sport organized activity by community trial participant characteristics.**

| Participant Characteristics | Odds Ratio | 95% Confidence Interval |
|---|---|---|
| Grade | | |
| 3rd | 0.92 | 0.34–2.49 |
| 4th | 0.96 | 0.33–2.85 |
| 5th | 1.62 | 0.63–4.20 |
| 6th | Reference (1.00) | |
| Sex | | |
| Male | 0.38* | 0.20–0.73 |
| Female | Reference (1.00) | |
| Family Income | | |
| Lower (Free/Reduced) | 0.55 | 0.26–1.16 |
| Higher (Full Pay) | Reference (1.00) | |

*$p < 0.05$

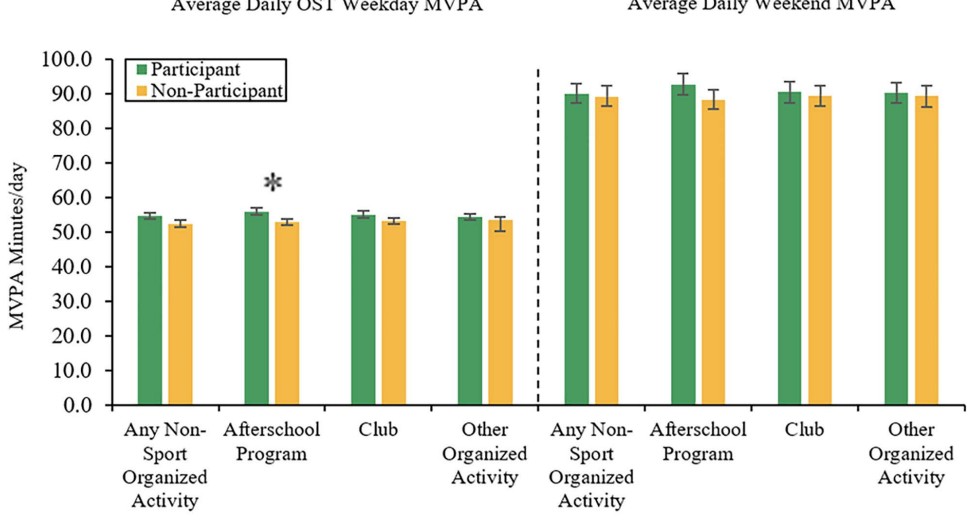

**Fig 1. Differences between out-of-school time (OST) organized activity participation and average daily minutes of OST moderate-to-vigorous physical activity (MVPA) per weekday and weekend.** Note that the error bars represent standard error; *$p < 0.05$.

those who did not participate in an afterschool program (mean = 53.1 ± 0.8 min/day). There were no significant differences in average minutes of weekend MVPA by afterschool program participation ($p = 0.060$). Additionally, there were no significant differences in average minutes of both weekday and weekend MVPA between participants and non-participants in clubs, other organized activities, and any OST non-sport organized activity.

### Weekday OST moderate-to-vigorous physical activity

Table 3 presents least squares means from the mixed-models and significant differences in daily OST MVPA per weekday by fixed effects of grade, sex, family income, and any OST non-sport organized activity participation. There were significant main effects on weekday OST MVPA for grade ($F(3,224) = 6.33$, $p < 0.001$) and sex ($F(1,224) = 114.88$, $p < 0.001$). Compared with 3rd, 4th, and 5th graders, 6th graders reported significantly less time in weekday OST MVPA. Additionally, males reported significantly more daily OST MVPA (mean = 61.0 ± 1.0 min/day) compared with females (mean = 46.2 ± 1.0 min/day). There were no significant main effects on weekday OST MVPA by family income ($F(1,224) = 1.46$, $p = 0.227$) and OST organized activity participation ($F(1,224) = 3.10$, $p = 0.080$).

There were, however, significant interaction effects on daily OST MVPA per weekday between family income and grade ($F(3,224) = 3.47$, $p = 0.017$) and family income and sex ($F(1,224) = 4.84$, $p = 0.029$), as also highlighted in Table 3. Fourth graders with lower income reported significantly fewer minutes of daily weekday OST MVPA than 4th graders with higher income (mean = 49.2 ± 2.8 and 57.9 ± 1.3, respectively). Conversely, 5th graders with lower income reported significantly more weekday OST MVPA (mean = 58.0 ± 1.9 min/day) than 6th graders with higher income (mean = 48.0 ± 1.4 min/day). In addition, males with lower income had significantly fewer minutes of daily OST MVPA per weekday than males with higher income (mean = 58.7 ± 1.8 and 63.4 ± 1.0, respectively). In contrast, females with lower income reported more minutes of daily OST MVPA (mean = 46.8 ± 1.6 min/day) compared with females with higher income (mean = 45.5 ± 1.1 min/day), although the difference was not significant.

### Weekend moderate-to-vigorous physical activity

Table 4 presents least squares means from the mixed-models and significant differences in average minutes per day of weekend MVPA by fixed effects of grade, sex, family income, and any OST non-sport organized activity participation, as well as the interaction effects. There was a significant main effect for grade ($F(3,226) = 27.27$, $p < 0.001$). Compared with 3rd, 4th, and 5th graders, 6th graders reported fewer minutes per day of weekend MVPA. Additionally, 5th graders reported significantly more daily minutes of weekend MVPA compared with 4th graders (mean = 98.4 ± 3.4 and 90.5 ± 3.5, respectively). There was also a significant main effect for sex ($F(1,226) = 65.45$, $p < 0.001$). Males reported significantly more weekend MVPA (mean = 100.5 ± 3.1 min/day) compared with females (mean = 79.0 ± 3.1 min/day). There were no significant main effects on weekend MVPA by family income ($F(1,227) = 1.16$, $p = 0.283$) and OST organized activity participation ($F(1,226) = 0.16$, $p = 0.687$).

Like weekday OST MVPA, there were significant interaction effects on weekend MVPA between family income and sex ($F(1,226) = 5.51$, $p = 0.020$), as highlighted in Table 4. Males with lower income had significantly fewer minutes of weekend MVPA (mean = 96.0 ± 4.2 min/day) than males with higher income (mean = 104.9 ± 3.1 min/day). In addition, females with lower income reported more daily minutes of weekend MVPA compared to females with higher income (mean = 80.6 ± 4.0 and 77.4 ± 3.2, respectively), although the difference was not significant. The interaction effect on weekend MVPA between family income and grade was not significant and was eliminated from the final model to enhance model fit.

### Discussion

The current study examined the influence of grade, sex, and family income on OST organized activity participation and their impact on rural children's OST MVPA during the weekday and weekend. In this study, community OST organized group activity participation included involvement in community settings (e.g., afterschool programs, clubs, organizations,

**Table 3. Least squares means estimates of average daily minutes of weekday OST MVPA by community trial participant characteristics.**

| Main Effects | MVPA, adjusted mean (95% CI) | Differences[a1] ($p < 0.05$) |
|---|---|---|
| Grade | | |
| a. 3rd | 54.6 (51.7–57.5) | d |
| b. 4th | 53.5 (50.5–56.6) | d |
| c. 5th | 56.9 (54.5–59.3) | d |
| d. 6th | 49.4 (46.8–52.0) | a, b, c |
| **Main Effects** | **MVPA, adjusted mean (95% CI)** | **Differences[b1] ($p < 0.05$)** |
| Sex | | |
| a. Male | 61.0 (59.0–63.1) | b |
| b. Female | 46.2 (44.2–48.1) | a |
| **Main Effects** | **MVPA, adjusted mean (95% CI)** | **Differences[c1] ($p < 0.05$)** |
| Family Income | | |
| a. Lower (Free/Reduced) | 52.8 (50.4–55.2) | None |
| b. Higher (Full Pay) | 54.4 (53.0–55.9) | None |
| **Main Effects** | **MVPA, adjusted mean (95% CI)** | **Differences[d1] ($p < 0.05$)** |
| Any OST Non-Sport Organized Activity Participation | | |
| a. Yes | 54.8 (53.2–56.3) | None |
| b. No | 52.4 (50.1–54.7) | None |
| **Interaction Effects** | **MVPA, adjusted mean (95% CI)** | **Differences[a2] ($p < 0.05$)** |
| Lower Family Income | | |
| a. 3rd | 53.2 (48.0–58.4) | None |
| b. 4th | 49.2 (43.7–54.6) | c, e, f, g |
| c. 5th | 58.0 (54.3–61.8) | b, d, h |
| d. 6th | 50.7 (46.2–55.1) | c, e, f |
| Higher Family Income | | |
| e. 3rd | 56.0 (53.4–58.6) | b, d, h |
| f. 4th | 57.9 (55.3–60.5) | b, d, h |
| g. 5th | 55.7 (52.8–58.7) | b, h |
| h. 6th | 48.0 (45.4–50.7) | c, e, f, g |
| **Interaction Effects** | **MVPA, adjusted mean (95% CI)** | **Differences[b2] ($p < 0.05$)** |
| Lower Family Income | | |
| a. Male | 58.7 (55.2–62.2) | b,c,d |
| b. Female | 46.8 (43.7–50.0) | a,c |
| Higher Family Income | | |
| c. Male | 63.4 (61.5–65.2) | a,b,d |
| d. Female | 45.5 (43.4–47.6) | a,c |

MVPA, moderate to vigorous physical activity; CI, confidence interval

[a1] Significance from mixed effects model (e.g., 'a' denotes difference from 3rd grade)

[b1] Significance from mixed effects model (e.g., 'a' denotes difference from male)

[c1] Significance from mixed effects model (no significant differences found)

[d1] Significance from mixed effects model (no significant differences found)

[a2] Significance from mixed effects model (e.g., 'a' denotes difference from 3rd grade with lower income)

[b2] Significance from mixed effects model (e.g., 'a' denotes difference from male with lower income)

**Table 4. Least squares means estimates of average minutes per day of weekend MVPA by community trial participant characteristics.**

| Main Effects | MVPA, adjusted mean (95% CI) | Differences[a1] ($p < 0.05$) |
|---|---|---|
| Grade | | |
| a. 3rd | 96.5 (84.3–108.6) | d |
| b. 4th | 90.5 (78.6–102.4) | c, d |
| c. 5th | 98.4 (85.9–110.9) | b, d |
| d. 6th | 73.5 (61.0–86.0) | a, b, c |
| **Main Effects** | **MVPA, adjusted mean (95% CI)** | **Differences[b1] ($p < 0.05$)** |
| Sex | | |
| a. Male | 100.5 (85.0–116.0) | b |
| b. Female | 79.0 (63.3–94.7) | a |
| **Main Effects** | **MVPA, adjusted mean (95% CI)** | **Differences[c1] ($p < 0.05$)** |
| Family Income | | |
| a. Lower (Free/Reduced) | 88.3 (75.5–101.1) | None |
| b. Higher (Full Pay) | 91.1 (67.2–115.1) | None |
| **Main Effects** | **MVPA, adjusted mean (95% CI)** | **Differences[d1] ($p < 0.05$)** |
| Any OST Non-Sport Organized Activity Participation | | |
| a. Yes | 90.2 (68.1–112.3) | None |
| b. No | 89.2 (76.0–102.4) | None |
| **Interaction Effects** | **MVPA, adjusted mean (95% CI)** | **Differences[a2] ($p < 0.05$)** |
| Lower Family Income | | |
| a. Male | 96.0 (85.5–106.5) | b,c,d |
| b. Female | 80.6 (69.9–91.3) | a,c |
| Higher Family Income | | |
| c. Male | 104.9 (88.5–121.4) | a,b,d |
| d. Female | 77.4 (63.0–91.8) | a,c |

MVPA, moderate to vigorous physical activity; CI, confidence interval

[a1] Significance from mixed effects model (e.g., 'a' denotes difference from 3rd grade)

[b1] Significance from mixed effects model (e.g., 'a' denotes difference from male)

[c1] Significance from mixed effects model (no significant differences found)

[d1] Significance from mixed effects model (no significant differences found)

[a2] Significance from mixed effects model (e.g., 'a' denotes difference from male with lower income)

etc.), excluding youth sports. A key finding was that approximately 70% of children reported participating in an OST organized activity within the past 12 months. Although evidence from other studies indicates availability and access to organized activities may be more limited in typical rural communities [25,70], our study suggests that within the participating rural communities, most children frequented these adult-led behavior settings. Sharp et al. [51] found similar results when examining organized activity participation among a cohort of predominantly non-Hispanic White adolescent students ($n = 276$) attending eight small public, rural schools in a remote county in the northeastern U.S. Specifically, the authors discovered the study's participants took advantage of existing community opportunities (e.g., OST organized activities like academic clubs and church youth groups) when school and community resources were limited (e.g., male team sports were less available and lack of funding to expand opportunity structures); plus, the patterns of participation remained stable as the students transitioned from middle to high school (i.e., across the 7th, 8th, and 10th grades) [51]. Thus, the effects of underpopulation (or "undermanning") witnessed by Barker and Gump in 1964 may still be relevant today given the high prevalence of participation among rural children and adolescents in these settings [71]. Even if rural children are

pressured to participate in OST organized activities out of necessity, participation can significantly impact childhood development, including the improvement of population health PA outcomes [38,72].

Characterizing children by grade level revealed no differences in OST organized activity participation, but sex was significantly associated with participation, with males being less likely to participate in OST organized activities compared to females. This finding is consistent with the literature [25,51,73]. Specifically, Guèvremont et al. [73] found that among Canadian children aged 6–17 years, both boys and girls were equally likely to participate in one organized extracurricular activity; however, a significantly ($p < 0.05$) larger proportion of girls than boys, across all age groups (i.e., 6–17 years), reported involvement in non-sport activities (e.g., music lessons, art, and drama) and clubs or community groups (e.g., 4-H, Scouts, and church groups). Some researchers may contend this difference exists because males are more likely to participate in youth sport opportunities compared to females [74,75], but with recent evidence suggesting females are just as, if not more, likely to participate in youth sports as males [46,76], future research should further investigate why the sex-based inequality in non-sport organized activity participation among rural children exists, explicitly if both non-sport and sport organized activity participation are becoming significantly higher among females than males.

Our work also revealed no differences in rural children's organized activity participation classified by family income. Although this finding may not be surprising in communities where "undermanning" exists and settings cannot afford to intentionally (or unintentionally) exclude children from participating, it is noteworthy since youth sport settings have been shown to marginalize children living in rural communities based on social classifications, including those from low-income families [46,47]. For example, Kellstedt et al. [46] found that FRLS was a significant ($p < 0.05$) predictor of youth sport participation, in which rural children from full pay families demonstrated almost four times higher participation rates in youth sport compared to those from FRLS families (OR = 3.91, 95% CI = 1.95–7.8). This suggests that OST non-sport organized activities may be more financially accessible than OST organized youth sport opportunities, particularly for economically disadvantaged children living in rural communities. Cost, however, is not the only factor influencing participation, and within rural communities, non-sport organized activities may be more responsive to different levels of environmental influences on children's behavior [30], such as parental or guardian need for childcare services after the school day (e.g., school-based afterschool programs) if they are commuting out-of-town for employment purposes or their employer does not offer flexible working hour or work-from-home options. Thus, it is necessary to account for different factors influencing OST organized activity participation, including who (e.g., parents, friends, and teachers) and what (e.g., availability, time constraints, and past experiences) are driving children's participation in these specific behavior settings. This information can uncover why children frequent specific community OST behavior settings within a given social context and could help identify ways to reverse or remove participation barriers and support more equitable organized activity opportunities where all children are encouraged to participate.

Although we found that children who participated in any OST organized activity were not significantly more active during OST on weekdays and weekends than nonparticipants, encouraging participation may still be beneficial given the contribution some OST organized activities have on children's PA, particularly when participating [77]. For instance, children in the present study who participated in afterschool programs reported significantly more daily minutes of OST MVPA per weekday ($mean_{diff} = 3.0 \pm 1.2$ min/day) than those who did not participate in an afterschool program ($p = 0.014$). Yet, there was no difference in weekend MVPA between program participants and non-participants, suggesting that afterschool programs may provide a meaningful contribution to PA, but only when children attend. Similarly, a systematic review and meta-analysis by Tassitano et al. [77] found that OST structured settings (i.e., afterschool programs and summer camps) provided youth (aged 3–18 years) with greater amounts of PA during attendance when compared to childcare and school, specifically, 11.7 min/hour and 6.4 min/hour, respectively. Since many (>70%) children in the U.S. are not physically active at the recommended levels [7,8], calls for any increase in childhood PA have been issued, as evidence suggests even small amounts or short episodes of PA provide health benefits both now and in the future (e.g., reduced anxiety symptoms and improved quality of life, respectively) [78–80]. Therefore, based on the results of our study and others examining the

PA patterns of children attending OST structured settings, like afterschool programs, if clubs and other organized activities can schedule time for PA within setting routines, this may positively influence PA promotion efforts. However, scholars warn simply allocating more time in the schedule for PA opportunities may not be sufficient for increasing children's MVPA. Specifically, the type of PA is also critical (e.g., structured vs. unstructured, individual vs. group) [81], especially in rural communities where evidence indicates non-sport organized activities are less structured and less supervised [25,38]. Thus, inserting shorter bouts of more structured PA opportunities into these settings may be more advantageous and practical for both rural children and adult leaders.

Another key finding in our study confirmed what is already known about the patterns of childhood PA characterized by grade and sex [12,82,83]. Specifically, we found significant differences in weekday OST and weekend MVPA by grade and sex. Compared with 3rd, 4th, and 5th graders, 6th graders reported the lowest average minutes of MVPA on both week-days and weekends. In addition, males reported higher amounts of MVPA than females on both weekdays and weekends (approximately 15 min/day and 21 min/day, respectively). Lower levels of MVPA as children age and lower levels of female MVPA have been presented in other studies, but reasons for these differences have been mixed [7,12,84]. A recently published 3-year cohort study objectively examined changes in MVPA of Slovenian schoolchildren while controlling for maturity and found a significant decrease in children's MVPA between ages 11- and 14-years regardless of pubertal timing [85]. Thus, the authors concluded that the drop in PA might not be biological but social structurally determined (e.g., reduced number of PE classes per week and more time spent completing schoolwork). Social structural differences, found within all layers of Bronfenbrenner's ecological systems theory from the immediate microsystem to the wider macrosystem [30], including broader societal gender norms influencing types of opportunities, roles within opportunities, and leader-level practices, may also help to explain sex-based PA disparities [54,86]. As a result, understanding the ecological processes and patterns underlying the formal and informal rules [87] driving PA behaviors not only between (e.g., entry rules and meeting space and time) but within (e.g., number of participants and session purpose) these settings may be critical in reducing PA-related health disparities.

Unlike grade and sex, family income did not significantly impact weekday and weekend OST MVPA. Scholars often postulate that children living in poverty are more likely to be less active than children from higher-income families [88–90]; however, evidence claims otherwise [91]. For instance, Whitt-Glover et al. [82] observed no differences in PA by socio-economic status among children aged 6–11 years when conducting a secondary analysis of national accelerometer data collected in 2003–2004. Voss et al. [92] also found that children from lower-income families were just as active as those from higher-income families, even though they attended significantly fewer sessions of organized activities.

Within the rural setting, Cottrell et al. [93] found parental support for children's PA was different based on family income. For example, parents with the lowest family income were more likely than parents from other income brackets to be more encouraging of PA and more directly involved (e.g., sending their children outside to play, praising them for being active, and discussing the importance of PA). In contrast, parents from the most affluent group were more likely to transport their children to be active than parents from all other income brackets [93]. Therefore, PA disparities based on family income might not be as prevalent in rural communities. Specifically, rural children from low-income families may be more active while outdoors in unstructured free play, thus, offsetting the PA accumulated by higher-income peers being transported to more structured activities [51,93]. Another possible explanation, consistent with Barker and colleagues' work on behavior settings, is that in often "undermanned" rural communities, children from lower-income families may feel obligated to participate in OST organized activities out of necessity or a desire for community belonging [71,94,95]; in turn, accumulating similar amounts of weekday and weekend OST MVPA with their higher-income participating peers. However, more research is warranted, particularly given the documented growing ethnoracial diversity in rural America over the past decade, marked by an influx of migrants and immigrants from Mexico and Latin America into communities with service and manufacturing opportunities (e.g., meat-processing plants), and the sug-gested decline in rural community pride [96,97], all of which may be contributing to changes in the local community

wellness landscape of organized settings and social norms. Therefore, the next step is to investigate the characteristics of rural OST organized activities, including the quality of programming, governance and organizational structure, and PA-promotion efforts, while considering parental support and transport patterns, explicitly within these so-called "new" rural American communities [98].

We also observed significant interactions between sex and family income for both primary outcomes of interest. Males with lower income reported significantly fewer minutes of OST weekday and weekend MVPA than males with higher income but were significantly more active than females regardless of family income. Although these findings are similar to those reported in other studies [99,100], some research contends males of lower socioeconomic status are more active than males of higher socioeconomic status [88,101]. Still, another study found the relation between socioeconomic status and sex was more pronounced among females, while there was no difference between males [89]. The so-called PA gender gap between boys and girls is well-established in the literature [102]; however, this gap may be overshadowing the fact that gaps within genders also exist and, thus, necessitate additional exploration.

Some evidence suggests that between- and within-gender gaps are further exacerbated in rural communities where male-dominated sports, including football and basketball, receive more support and attention, thus, encouraging males to obtain greater social status by adopting a "jock identity" [25,103]. This social identity, often associated with an ego-oriented approach to sports that transcends beyond the playing field and into other aspects of life (e.g., school classroom or the dinner table) [104], may help to explain why we found that males were more active than females regardless of family income; however, it does little to dissect the within-gender gap between males. Hankonen et al.'s [105] attempt at explaining the socioeconomic status gap in activity among Finnish vocational and high school students ($n=659$) aged 16–19 years is worth noting, as even though the authors highlighted that self-efficacy was highest among high schools boys (high socioeconomic status group) and lowest among vocational boys (low socioeconomic status group) ($p=0.039$), they acknowledged the need to examine how long-term, societal level processes might be causing differences. Hence, within-gender gaps, such as the PA gap between rural boys from different socioeconomic backgrounds characterized in our study, are worth investigating.

## Limitations

Provided the diversity of OST organized activities within local communities and across the U.S. and the fact our sample was limited to two communities and their respective organized activities, the results of this study should be interpreted in light of its limitations. The residents in the communities from this wave of the Wellscapes Project were primarily non-Hispanic White, so we were unable to examine PA disparities by race and ethnicity nor explore the association between these factors and participation in OST organized activities. However, work with more diverse communities (with a concentration of Hispanic residents) in Wave 2 will allow for a deeper investigation of ethnoracial PA and participation differences. Additionally, future research should continue investigating the influence of family income on non-sport organized activity participation within a rural context, as even though we found no significant difference in the likelihood of participating based on family income, we acknowledge the limited generalizability of this and other study findings given the presence of sampling and selection bias. For example, most children (75.3%) in the community trial were from higher-income families, although this percentage was relatively consistent with first-year total school enrollment information presented by Schenkelberg et al. [62], in which 69.5% of enrolled children were from higher-income families. We acknowledge our study's definition of family income based on lunch status also warrants attention, as a more comprehensive indicator, like social economic status, may be more precise [106]. However, this classification is considered a sufficient proxy given the privacy requirements associated with school student enrollment records [66].

Furthermore, we acknowledge the limitation of the cross-sectional study design, in which a causal link between participation in OST non-sport organized activities, including daily afterschool programs, and OST MVPA cannot be established. There are also limitations to the self-report measure of OST organized activity involvement and PA in the present study,

even though the YAP is designed to provide reasonably accurate group-level estimates of in-school and out-of-school PA. Comparison of participation results can occur across national, state, and local levels with the inclusion of the NSCH questions, but future research should also consider employing geographic information system (GIS) and other mapping tools to gain more objective data about the participation in OST organized activities and the places children frequent in and outside rural communities, including locations where unstructured activities, like free play, occur. In addition, the inclusion of an objective PA outcome measure (e.g., accelerometry) would also strengthen the study, but data collection efforts are resource-intensive [26]. Thus, improving the self-report instrument to be more sensitive to the amount of time children spend in each organized activity and the frequency of the activity, impacting OST weekday and weekend MVPA, may be a critical first step [51]. It would also be beneficial to examine children's participation and PA trends over time to inform data-driven decision-making at multiple levels (e.g., setting and community levels).

## Conclusions

The present study aimed to examine the sociodemographic factors related to out-of-school time (OST) non-sport organized activity participation and how these factors interact to influence OST physical activity (PA) among children living in rural U.S. communities. The findings suggest that, along with other factors, overall OST non-sport organized activity participation does not play a critical role in maximizing the amount of PA that rural children engage in during OST on weekdays and weekends. Yet, these settings should not be overlooked in PA-promotion efforts provided the fairly high percentage (70%) of rural children participating in non-sport organized activities, coupled with the fact that grade and family income were not a significant driver of participation in this study. Thus, non-sport OST organized activities may be more accessible and responsive to different levels of environmental influences on rural children's behavior, particularly when considering the well-documented barriers to organized youth sport participation, like cost. More research is needed to explore how these activities could be effectively designed to increase PA since "some physical activity is better than none" [79] and afterschool programs, which often devote time to PA within daily setting routines, can provide a blueprint to consider. Therefore, increasing participation and opportunities for PA within non-sport organized activities could positively impact the MVPA and population health outcomes of children living in rural communities, and advocating for the advancement of an updated National Youth Sports Strategy [42] to include non-sport organized activities may aid in ensuring children nationwide have the opportunity to achieve optimal health.

## Acknowledgments

The authors would like to thank all community members contributing to this study and Sara J. Norgelas, MS, for her involvement in coordinating data collections.

## Author contributions

**Conceptualization:** Mary J. Von Seggern, Michaela A. Schenkelberg, Ann E. Rogers, Debra K. Kellstedt, Gregory J. Welk, Richard R. Rosenkranz, David A. Dzewaltowski.

**Formal analysis:** Ann E. Rogers, Robin High, David A. Dzewaltowski.

**Funding acquisition:** David A. Dzewaltowski.

**Investigation:** Mary J. Von Seggern, Ann E. Rogers, Gregory J. Welk, David A. Dzewaltowski.

**Supervision:** David A. Dzewaltowski.

**Writing – original draft:** Mary J. Von Seggern.

**Writing – review & editing:** Michaela A. Schenkelberg, Ann E. Rogers, Debra K. Kellstedt, Robin High, Gregory J. Welk, Richard R. Rosenkranz, David A. Dzewaltowski.

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
