## [Decision Letter · Decision Letter 0]

20 Aug 2024

PONE-D-24-23961The association between children’s participation in out-of-school time organized activities and physical activity in rural communities: A cross-sectional studyPLOS ONE

Dear Dr. Von Seggern,

Thank you for submitting your manuscript to PLOS ONE. After careful consideration, we feel that it has merit but does not fully meet PLOS ONE’s publication criteria as it currently stands. Therefore, we invite you to submit a revised version of the manuscript that addresses the points raised during the review process. We believe the reviewers' comments can help you improve your manuscript significantly, and you might refer to some editor comments below.  

We look forward to receiving your revised manuscript.

Kind regards,

Catherine M. Capio

Academic Editor

PLOS ONE

 [This work was supported by the National Cancer Institute of the National Institutes of Health under Award Number R01CA215420.  Study sponsors were not involved with data collection, analysis, interpretation, or writing of the manuscript.  The content is solely the responsibility of the authors and does not necessarily represent the official views of the National Institutes of Health. ].  

Additional Editor Comments:

We received two reviews of your manuscript with different recommendations. Having reviewed the comments and your manuscript myself, I believe the concerns raised may be addressed by a thorough revision process which could ultimately improve the manuscript. As such, I invite you to consider the reviewers' comments and submit a revision. In particular, I tend to agree with the reviewers that there are aspects in the paper that need to speak to a more global audience that may not be familiar with the context in your region. You might also consider acknowledging limitations with a view to what might be done in the next stage to address those limitations, besides providing the justifications for those limitations.

Reviewers' comments:

Reviewer's Responses to Questions

**Comments to the Author**

1. Is the manuscript technically sound, and do the data support the conclusions?

Reviewer #1: Yes

Reviewer #2: Yes

2. Has the statistical analysis been performed appropriately and rigorously? 

Reviewer #1: I Don't Know

Reviewer #2: Yes

3. Have the authors made all data underlying the findings in their manuscript fully available?

Reviewer #1: No

Reviewer #2: Yes

4. Is the manuscript presented in an intelligible fashion and written in standard English?

Reviewer #1: Yes

Reviewer #2: Yes

5. Review Comments to the Author

Reviewer #1: Thank you very much for the opportunity to review this manuscript. For ethical reasons, I have chosen to introduce myself. My name is Johan Högman, and I work at Karlstad University in Sweden. I agreed to review this manuscript due to its focus on physical activity among rural children outside of organised sports—a topic I believe is of great importance.

I appreciate the clear structure and overall clarity of the manuscript. This includes the methodological rigour as well as the presentation of the findings. It is a commendable achievement to collect this type of data, and I wish to congratulate the authors for this effort. I also value your focus on the contribution of non-sport activities, as this is highly significant for physical activity research. I believe this paper will make an important contribution to the field.

I have a few comments that I have listed below.

Introduction

- The introduction provides a good overview of the current state of research and the problem formulation. The depiction of differences in physical activity between urban and rural populations is nuanced.

- One area for development in the introduction is the study's justification. For example, lines 106-113 attempt to justify the study but without drawing a conclusion. Why is it important to study the relationship between different physical activity settings? And how is this addressed in this study?

- Lines 132-140 also contribute well to the study’s relevance. However, I suggest that this discussion be expanded by adding how the knowledge from this study will contribute to the development of policies and practices.

- Line 209 – I suggest explaining how the school lunch system operates in the United States, as this is not something that functions the same way globally.

- I also suggest that the subject of the study—OST activities—be described in more detail. For example, nothing is said about the costs associated with participating in these activities, which is relevant since family finances are one of the factors being examined.

Methods

- Lines 269-271 – It is somewhat confusing to follow how the two different samples are used. If I understand correctly, the difference is only that in the "epidemiological" study, there is no access to data on lunch status and race/ethnicity (?) So, when results are presented without mentioning which sample is being used (e.g., in Table 2), should one assume it is the "epidemiological" study with 435 individuals? However, lines 187-190 state that “This study primarily focused on a subset of those children (n = 235) who had informed parental and guardian written consent to participate in the Wellscapes community randomised trial and pair children’s school sociodemographic data.” I suggest, as mentioned, that it be clarified throughout the paper how these samples were used for the different calculations.

Discussion

- Lines 417-424 – I am uncertain whether it is relevant to refer to studies that include sports activities when arguing for the value of non-sport activities. To me, this is not entirely convincing.

- Lines 484-488 – This may be understood by Americans but is difficult to relate to for someone like me, a European. Please clarify what is meant by the discussion regarding the "new" rural America. Sounds interesting!

- Lines 501-502 – In the same vein as the previous comment: I recommend adding a sentence to explain the concept of a “jock,” as it may not be as widely recognised outside of North America.

- Lines 542-544 – This sentence is not entirely clear and needs clarification.

- I wonder whether the dropout rate in the "trial study" needs to be addressed in the limitations. Could it be that those who did not consent to provide data on lunch status were more likely to have a lower socioeconomic status (which is often the case in dropout situations)? If so, it is likely that this has influenced the results regarding who participates in OST activities.

Conclusions

- I also wonder whether the study's conclusions and contributions to the research field could be further clarified by being more explicitly related to the ecological theories briefly mentioned at the beginning of the paper (Bronfenbrenner/Barker). Is it possible to understand why the behaviour (PA) is not transferred from OST activities to weekend MVPA, or perhaps why the participation rate is as high as 70% in OST activities, using these theories? A clearer contextualisation and theoretical positioning at the end would enhance the paper’s significance.

- I would appreciate it if the measurement instrument were included as a supplementary file upon publication, though I am not entirely sure of the journal's guidelines.

Reviewer #2: Thank you for the opportunity to review this paper. This paper examined the associations of demographic factors with non-sport organized activity participation and OST PA in children from a rural area. I have a few comments for the authors to consider:

1. line 142, the association between participation in community non-sport OST and?

2. Line 254-262 Please provide more details on how OST MVPA and weekend MVPA were calculated in minutes from these YAP items?

3. Line 282 for weekday, was this OST MVPA?

4. It would be beneficial to present the interaction findings in figures.

5. While some statistically significant differences were observed in this study, these differences were 3 minutes over a half day on weekdays or less than 10 minutes over a whole day – which may be less meaningful in terms of the health impact. How would the authors interpret the clinical significance of their findings?

6. The title seems not accurately reflect the study aims which examined the associations of a few factors with non-sport organized activity participation and OST PA in children from a rural area.

7. While the authors highlight the limited research on children's physical activity (PA) in rural areas compared to urban ones, the significance and novelty of the present study remain unclear. The study relied on self-reported questionnaires, which may not provide a comprehensive understanding of PA patterns in the targeted sample. Could the authors further elaborate on what makes this study novel and important?

6. PLOS authors have the option to publish the peer review history of their article (what does this mean? ). If published, this will include your full peer review and any attached files.

**Do you want your identity to be public for this peer review?** For information about this choice, including consent withdrawal, please see our Privacy Policy .

Reviewer #1: **Yes: ** Johan Högman

Reviewer #2: No

---

## [Author Response · Author response to Decision Letter 1]

18 Oct 2024

Dear Dr. Capio and Reviewers (including Dr. Högman),

On behalf of my colleagues and myself, thank you for inviting us to submit a revised version of our manuscript, originally titled, “The association between children’s participation in out-of-school time organized activities and physical activity in rural communities: A cross-sectional study,” to PLOS ONE. We also appreciate the time and effort each of you have dedicated to providing valuable feedback on ways to strengthen our manuscript. As a result, we have incorporated changes in the revised manuscript that reflect the detailed suggestions you all have provided in the letter dated August 20, 2024. We have attempted to succinctly explain changes made in reaction to all comments in a point-by-point fashion displayed in a table format in our Cover Letter/Rebuttal Letter, corresponding with the letter’s organization (i.e., Journal Requirements and Prompts, Additional Editor Comments, and Reviewers’ Comments). Therefore, we suggest referring to the Cover Letter/Rebuttal Letter with Response to Reviewers document, as the table format provides clarity matching Editor and Reviewer comments with the corresponding responses. However, we have pasted our specific responses below (not including the table format due to the text box configurations).

We also hope that our edits and the responses we have provided below satisfactorily address all the issues and concerns everyone has noted.

Again, thank you for giving us the opportunity to strengthen our manuscript with the valuable comments and queries, and we have worked hard to incorporate all feedback. We look forward to hearing from you regarding our submission and would be glad to respond to any further questions and comments that any of you may have.

Sincerely,

Mary J. Von Seggern

Editor Comments

Author Response and Modification

1. Thank you for this feedback, and we have revised aspects of the paper to speak to a more global audience, explicitly related to the feedback provided by the reviewers. Our responses are highlighted in Part III, including adding details about the U.S.’s lunch system/status (Reviewer #1, Comment 4), “new” rural America (Reviewer #1, Comment 8), and “jock identity” (Reviewer #1, Comment 9). Additional information was also added throughout the manuscript to provide more clarity, consistent with the comments shared by Reviewer #2.

2. We have revised the Limitations section based on your feedback, as well as the reviewers’ feedback, to highlight the following:

2.1 First, we added a Limitations subheading in the Discussion section (Page 27, line 577) to clearly delineate this information.

2.2 Secondly, we updated the Limitations section to include the following:

Provided the diversity of OST organized activities within local communities and across the U.S. and the fact our sample was limited to two communities and their respective organized activities, the results of this study should be interpreted in light of its limitations. The residents in the communities from this wave of the Wellscapes project were primarily non-Hispanic White, so we were unable to examine PA disparities by race and ethnicity nor explore the association between these factors and participation in OST organized activities. However, work with more diverse communities (with a concentration of Hispanic residents) in Wave 2, will allow for a deeper investigation of ethnoracial PA and participation differences. Additionally, future research should continue investigating the influence of family income on non-sport organized activity participation within a rural context, as even though we found no significant difference in the likelihood of participating based on family income, we acknowledge the limited generalizability of this and other study findings given the presence of sampling and selection bias. For example, most children (75.3%) in the community trial were from higher-income families, although this percentage was relatively consistent with first-year total school enrollment information presented by Schenkelberg et al. (2021), in which 69.5% of enrolled children were from higher-income families [Schenkelberg et al. 2021]. We acknowledge our study’s definition of family income based on lunch status also warrants attention, as a more comprehensive indicator, like social economic status, may be more precise [96]. However, this classification is considered a sufficient proxy given the privacy requirements associated with school student enrollment records [52]. (Page 28, Lines 584-597)

2.3 We also added more information to the second paragraph in the Limitations section:

Next, there are limitations to the self-report measure of OST organized activity involvement and PA in the present study, even though the YAP is designed to provide reasonably accurate group-level estimates of in-school and out-of-school PA. Comparison of participation results can occur across national, state, and local levels with the inclusion of the NSCH questions, but future research should also consider employing geographic information system (GIS) and other mapping tools to gain more objective data about the participation in OST organized activities and the places children frequent in and outside rural communities, including locations where unstructured activities, like free play, occur. In addition, the inclusion of an objective PA outcome measure (e.g., accelerometry) would also strengthen the study, but data collection efforts are resource-intensive (Essay et al. 2023). Thus, improving the self-report instrument to be more sensitive to the amount of time children spend in each organized activity and the frequency of the activity, impacting OST weekday and weekend MVPA, may be a critical first step [44]. It would also be beneficial to examine children’s participation and PA trends over time to inform data-driven decision-making at multiple levels (e.g., setting and community levels). (Page 29, Lines 605-620)

Reviewers’ Comments

Author Response and Modification

Reviewer #1

1. We appreciate your feedback and agree the depiction of differences in physical activity between urban and rural populations is complex, further complicated by differences between and within communities and the behavior settings nested within.

2. We have revised the aforementioned paragraph in the Introduction to include the following information and further develop the study’s justification:

Building upon Barker’s definition of communities as systems of behavior settings, in which ecological units bounded by time and space drive patterns of behavior within a replicated social structure [24], the concern here is to forgo the classic rural-urban comparison and examine the population PA outcomes of children residing within rural communities. This approach emphasizes the need to move beyond the oversimplified rural typology and embrace the complexity of rural communities and the system of organized settings (e.g., school classrooms, youth sport practices, and club meetings) nested within, particularly given the variability of available and accessible settings across rural communities [Dzewaltowski 2017; Ferris et al. 2013]. Population PA outcomes for children are highly dependent on the interactions among individuals within setting environments (e.g., peer influence and leader behavior) and among the community “wellness landscape” [25] of organized settings children frequent, including school and out-of-school time (OST) organized settings, with different structuring properties (i.e., rules and resources) [Schlechter et al. 2017; Li and Moosbrugger 2021; Giddens 1984]. Bronfenbrenner’s ecological theory [28] further aligns with this approach and the need to study the complex interplay between children, their immediate environments, and the larger whole-of-community system [29]; however, much of the literature has examined children’s PA behaviors solely within the school setting, often isolating individual outcomes and neglecting the places children go during OST [Stanley et al. 2012; Messing et al. 2019]. Thus, investigating the places rural children frequent outside of the school day is critical to understanding their PA behaviors. (Pages 4-5, lines 109-127)

3. We revised this section as follows:

Organized youth sport, where PA is a primary purpose, has been identified as an effective public health strategy to increase childhood PA [37], but the current pay-to-play model with an estimated average $883 annual price tag per child and sport (Aspen Institute Project Play 2022], among other factors (e.g., overemphasis on sport specialization and winning-at-all-costs), has erected barriers to participation [38,39]. For example, the presence of youth sport participation barriers among children living in rural Great Plains communities was recently highlighted by Kellstedt and colleagues (2021) and Von Seggern et al. (2024), in which children from higher-income families were almost four times more likely to participate in youth sport than their lower-income peers [Kellstedt et al. 2021], and non-Hispanic White children were over five times more likely to participate than Hispanic children [Von Seggern et al. 2024], respectively. Thus, understanding if non-sport OST organized opportunities are accessible, regardless of primary purpose, is fundamental as these often-overlooked opportunities with substantial cost variability (e.g., within Scouting, cost estimates range anywhere from $110 to over $600 annually per child) [ScoutSmarts 2024] may be critical to reaching more children for PA-promotion efforts, including those priced out of youth sports.

Despite the potential reach of these settings, the contribution of non-sport organized activity involvement, such as participating in clubs and youth organizations (e.g., 4-H, Scouting, and STEM), to children’s PA is not well understood [Virgara et al. 2021; Brooke et al. 2014]. Further, even less is known about non-sport OST organized activity participation and PA among rural children, although patterns of involvement in these activities are beginning to emerge in the literature [27,43,44]. Evidence suggests settings where PA is not the primary purpose (e.g., schools and afterschool programs) can improve children’s PA by inserting opportunities for PA (e.g., physical education classes, outdoor play, and brain breaks) or adopting PA policies, like the National AfterSchool Association’s (2018) healthy eating and physical activity (HEPA) standards or other active recreation policies [Beets et al. 2009; Schlechter et al. 2018; Woods et al. 2021]. However, Sliwa et al. (2019) acknowledged that many OST organized opportunities lack PA policies and practices or awareness of existing policies, particularly in those administered outside of schools and school districts [Sliwa et al. 2019]. Therefore, understanding the places children go outside of the school day, the quantity of the PA provided by these opportunities, and how these and other factors (e.g., sociodemographic characteristics) interact to influence childhood PA outcomes, will help to inform non-sport setting design and better support the integration of public health policies, systems, and environmental change strategies related to PA-promotion efforts, explicitly in understudied and underserved rural communities. (Pages 5-7, lines 143-182)

4. Thanks for bringing this to our attention, and we have revised this section in the Methods by adding more information about lunch status as follows:

Individual sociodemographic characteristics in this study included grade, sex, and a proxy for family income (i.e., lunch status) based on student eligibility requirements for free or low-cost meals during the school day (e.g., children from families with incomes at or below 130% or between 130% and 185% of the federal poverty line, respectively) as part of the National School Lunch Program [U.S. Department of Agriculture 2024], a U.S. Department of Agriculture federally assisted meal program operating in public and private schools and residential child care institutions across the nation [Sinclair and Chen 2020]. (Page 11, lines 256-261)

5. Thanks for suggesting this, and although there is substantial cost variability with OST activities, we have revised several paragraphs in the Introduction to include the following:

5.1 However, OST organized activities vary widely within local communities and across the U.S., serve many different purposes (e.g., academic, youth development, enrichment, or childcare), and rely on different funding mechanisms (e.g., public or private sources) to cover programming costs, not limited to staff time or necessary equipment [Grossman et al. 2009]. (Page 5, lines 138-141)

5.2 Organized youth sport, where PA is a primary purpose, has been identified as an effective public health strategy to increase childhood PA [37], but the current pay-to-play model with an estimated average $883 annual price tag per child and sport (Aspen Institute Project Play 2022], among other factors (e.g., overemphasis on sport specialization and winning-at-all-costs), has erected barriers to participation [38,39]. For example, the presence of youth sport participation barriers among children living in rural Great Plains communities was recently highlighted by Kellstedt and colleagues (2021) and Von Seggern et al. (2024), in which children from higher-income families were almost four times more likely to participate in youth sport than their lower-income peers [Kellstedt et al. 2021], and non-Hispanic White children were over five times more likely to participate than Hispanic children [Von Seggern et al. 2024], respectively. Thus, understanding if non-sport OST organized opportunities are accessible, regardless of primary purpose, is fundamental as these often-overlooked opportunities with substantial cost variability (e.g., within Scouting, cost estimates range anywhere from $110 to over $600 annually per child) [ScoutSmarts 2024] may be critical to reaching more children for PA-promotion efforts, including those priced out of youth sports. (Pages 5-6, lines 143-157)

6. Thanks for sharing this with us, and we have attempted to provide additional clarification about which sample is being used (in relation to the social epidemiology study and community trial study) in the following paragraph:

6.1 Descriptive statistics were used to summarize children’s participation in organized activities for the social epidemiology study (n = 418) and the subset of children included in the community trial (n = 235). Mixed-models were used to analyze the dichotomous outcomes (i.e., 1 = Yes; 0 = No) of any OST organized activity, afterschool program, club, and other organized activity (e.g., music and dance) participation, and the continuous outcomes of average daily minutes of OST MVPA on the weekdays and average daily minutes of MVPA on weekend days for community trial participants only. (Page 14, Lines 327-332)

We also added some additional information in the Results section for more clarity:

6.2 Table 2 shows the results from the mixed-model regression predicting OST organized activity participation among community trial participants. (Page 16, Line 361)

6.3 Added “community trial” participant characteristics in Table 2, Table 3, and Table 4 headings. (Pages 16, 18, and 20, respectively)

6.4 Added “among community trial participants” when referencing Fig 1. (Page 17, line 366)

7. We value this feedback and have removed the inclusion of the ‘PA/sport programs’ information. However, we feel strongly about including the reference (by Tassitano and colleagues, 2020), since this is one of the first attempts to conduct a systematic review and meta-analysis examining the amount of time youth spend being physically active in structured settings. Additionally, the authors’ definitions of afterschool programs and summer day camps share similarities with the non-sport-organized activities we examined in our study. We could have selected one of the studies reviewed to support our claims; however, including this reference allows the reader to consider multiple studies. Plus, we highlighted one specific study by Trost et al. (2008) following the inclusion of Tassita

---

## [Decision Letter · Decision Letter 1]

16 Dec 2024

PONE-D-24-23961R1Sociodemographic influences on children’s out-of-school time organized activity participation and physical activity in rural communities: A cross-sectional studyPLOS ONE

Dear Dr. Von Seggern,

Thank you for submitting your manuscript to PLOS ONE. After careful consideration, we feel that it has merit but does not fully meet PLOS ONE’s publication criteria as it currently stands. Therefore, we invite you to submit a revised version of the manuscript that addresses the points raised during the review process.

Thank you for your revision. Please have a look at a number of remaining concerns, which could potentially improve the dissemination of knowledge through your manuscript. Please submit your revised manuscript by Jan 30 2025 11:59PM. If you will need more time than this to complete your revisions, please reply to this message or contact the journal office at plosone@plos.org . Please include the following items when submitting your revised manuscript:

We look forward to receiving your revised manuscript.

Kind regards,

Catherine M. Capio

Academic Editor

PLOS ONE

Journal Requirements:

Additional Editor Comments:

Thank you for your revised manuscript. While the points raised in the original submission were generally addressed, there are a few outstanding concerns that the authors should consider. I also note that the comment from Reviewer #2 from the previous version might have been a bit misunderstood; consider re-thinking the revision related to this.

Reviewers' comments:

Reviewer's Responses to Questions

**Comments to the Author**

1. If the authors have adequately addressed your comments raised in a previous round of review and you feel that this manuscript is now acceptable for publication, you may indicate that here to bypass the “Comments to the Author” section, enter your conflict of interest statement in the “Confidential to Editor” section, and submit your "Accept" recommendation.

Reviewer #1: (No Response)

Reviewer #2: (No Response)

2. Is the manuscript technically sound, and do the data support the conclusions?

Reviewer #1: Yes

Reviewer #2: Yes

3. Has the statistical analysis been performed appropriately and rigorously? 

Reviewer #1: I Don't Know

Reviewer #2: Yes

4. Have the authors made all data underlying the findings in their manuscript fully available?

Reviewer #1: No

Reviewer #2: Yes

5. Is the manuscript presented in an intelligible fashion and written in standard English?

Reviewer #1: Yes

Reviewer #2: Yes

6. Review Comments to the Author

Reviewer #1: Thank you for the opportunity to review this revised manuscript. I would like to commend the authors for a structured and thoroughly conducted revision. I believe most of my previous comments have been addressed, and in many cases, clarified effectively. I have a few further comments on this revised version.

- Use of Theory: Barker/Bronfenbrenner is employed as a form of justification for the study – a purpose I find less convincing unless the authors directly tie back to this framework later in the paper. In the discussion, other, more loosely connected explanatory models are introduced instead. I would still suggest that the findings be discussed in relation to the theories highlighted in the introduction. If not, it may be better to remove these theoretical references from the introduction altogether to avoid confusion.

Results/Discussion

- The presentation of results related to the contribution of OST activities to PA is somewhat confusing. You found no significant differences between participants and non-participants in organised OST activities, yet you suggest (p. 22, line 450) that these activities can still be beneficial for PA. This appears contradictory and should be reviewed.

- In the discussion, there is an unacknowledged shift from OST organised activities to afterschool activities. If you intend to discuss afterschool programmes and present findings specific to them, the relationship between these types of activities should be described and defined more clearly earlier in the paper. While this is addressed to some extent, I believe clearer definitions would aid an international readership in understanding the results. Additionally, the rationale for this shift in focus should be clarified – I suggest explaining why the discussion transitions from OST in general to afterschool programmes specifically.

- Conclusions: I believe the Conclusions section would benefit from further revision to contextualise the study’s key findings. If the authors do not wish to tie back to the theoretical framework, the key findings should at least be highlighted more clearly. Currently, much of the Conclusions section consists of repetition.

Minor Comments

- Reference formatting does not consistently follow the Vancouver style throughout the text.

I hope my comments are seen as constructive and can help to further develop and clarify the important contributions of the article.

Reviewer #2: Thank you for the opportunity to review this paper again. It would be helpful if the authors could refer to the comments in their response.

-While the authors have addressed most my concerns, I believe my original comment #5 was not fully addressed. My question was not about the significance of this study but the practical significance of the study. While the observed group level differences were statistically significant, the differences were 3 minutes over a half day on weekdays or less than 10 minute over a whole day. Are these differences meaningful in terms of the health impact? how do the authors interpret the practical significance?

-The revised conclusion section is too lengthy. I recommend transferring some of the content to the discussion section.

7. PLOS authors have the option to publish the peer review history of their article (what does this mean? ). If published, this will include your full peer review and any attached files.

**Do you want your identity to be public for this peer review?** For information about this choice, including consent withdrawal, please see our Privacy Policy .

Reviewer #1: **Yes: ** Johan Högman

Reviewer #2: No

---

## [Author Response · Author response to Decision Letter 2]

27 Jan 2025

Catherine M. Capio, Ph.D.

Academic Editor

PLOS ONE

January 27, 2025

Dear Dr. Capio,

Re: Resubmission of manuscript reference No. PONE-D-24-23961R1

On behalf of my colleagues and myself, thank you for inviting us to address some remaining concerns and submit a revised version of our manuscript, now titled, “Sociodemographic influences on children’s out-of-school time organized activity participation and physical activity in rural communities: A cross-sectional study,” to PLOS ONE. We also appreciate the time and effort you and each of the reviewers have dedicated to providing additional feedback on ways to improve the dissemination of knowledge through our manuscript. As a result, we have incorporated changes in the revised manuscript that reflect the comments you and the reviewers have provided in your letter dated December 16, 2024. Similar to our previous revision, we have attempted to explain changes made in reaction to all comments in a point-by-point fashion displayed in a table format in the attached "Response to Reviewers" document, corresponding with the letter’s organization (i.e., Journal Requirements, Additional Editor Comments, and Reviewers’ Comments), and hope that our edits and the responses we have provided below satisfactorily address all the remaining concerns you and the reviewers have noted.

Again, thank you for giving us the opportunity to strengthen our manuscript with your valuable feedback. We look forward to hearing from you regarding our submission and would be glad to respond to any further questions and comments that you and the reviewers may have.

Sincerely,

Mary J. Von Seggern

Part I – Journal Requirements

Comments

Journal Requirements – Reference List

Author Response and Modification:

1. Thank you for bringing this to our attention, and we have reviewed our reference list to ensure it is complete and correct, including based on the changes listed below (in which a total of 7 references were completely removed from the paper and 5 references were added):

1.1 We removed references 76-78 (Belcher et al., 2010; Change et al., 2019; and Whitt-Glover et al., 2009) on pg. 22, line 440, since we revised this paragraph based on Reviewer #1’s feedback of discussing findings in relation to the ecological theories presented in the beginning; although, the Whitt-Glover et al., 2009 citation was used later in the paper (so it is still in the reference list).

1.2 Additionally, we removed reference 79 (Probst et al., 2018) on pg. 22, line 446, and reference 80 (Yousefian et al. 2009) on pg. 22, line 453, since we are no longer comparing rural children to urban peers (and focusing on differences within rural community systems and connecting the findings back to ecological theories presented in the beginning).

1.3 To further address Reviewer comments in the Discussion section, primarily related to the unacknowledged shift from OST organized activities to afterschool programs, as well as the need to address the practical significance of the study, we removed afterschool references, specifically reference 82 (Trost et al., 2008) on pg. 23, line 480 and reference 83 (Mears and Jago, 2016) on pg. 24, line 488. Although reference 84 (Weaver et al., 2015) was deleted on pg. 24, line 490, it was not fully removed from the article as it was cited on pg. 24, line 496.

1.4 To support the practical significance of the study, we added three references, specifically: (1) Piercy et al., 2018; (2) Piercy and Troiano, 2018; and (3) Sriram et al., 2021, on pg. 23, lines 477-478.

1.5 We also added two citations, Barker and Schoggen, 1973, and Gump and Adelberg, 1978, on pg. 26, line 544, to support one of our study’s finding related to family income and weekday and weekend OST MVPA. In response to this change (and emphasis on Barker and colleagues’ work to support the finding), we removed reference 96 (Ludden, 2011) and reference 25 (Ferris et al., 2013), but the latter reference remains included in the article from previous citations.

Part II – Additional Editor Comments

Comments

Editor: General Comments

Thank you for your revised manuscript. While the points raised in the original submission were generally addressed, there are a few outstanding concerns that the authors should consider.

Author Response and Modification:

Thank you for taking time to review and provide comments on our revised manuscript. We appreciate your feedback, as well as the opportunity to revise and resubmit the manuscript once again.

Editor: Particular Comments

1. I also note that the comment from Reviewer #2 from the previous version might have been a bit misunderstood; consider re-thinking the revision related to this.

Author Response and Modification:

1. Thank you for this feedback, and we realize the comment from Reviewer #2 from the previous version was misunderstood and have revised the Discussion to include a specific statement related to the study’s practical significance as follows:

Since many (>70%) children in the U.S. are not physically active at the recommended levels [7,8], calls for any increase in childhood PA have been issued, as evidence suggests even small amounts or short episodes of PA provide health benefits both now and in the future (e.g., reduced anxiety symptoms and improved quality of life, respectively) [Piercy et al. 2018; Piercy and Troiano 2018; Sriram et al. 2021]. Therefore, based on the results of our study and others examining the PA patterns of children attending OST structured settings, like afterschool programs, if clubs and other organized activities can schedule time for PA within setting routines, this may positively influence PA promotion efforts (Pages 23-24, lines 473-493)

We also added a comment in the Conclusions section about the practical significance as follows:

From a public health perspective, advocates acknowledge “some physical activity is better than none” [Piercy and Troiano 2018]; therefore, it might be worthwhile for the U.S. to expand the National Youth Sports Strategy [U.S. Department of Health and Human Services 2019] to include non-sport organized activities. (Page 31, lines 641-644)

2. We also revised the Conclusions section based on both Reviewers’ feedback (highlighted below) and addressed all other comments accordingly.

Part III – Reviewers’ Comments

Comments

Reviewer #1: General Comments

Thank you for the opportunity to review this revised manuscript. I would like to commend the authors for a structured and thoroughly conducted revision. I believe most of my previous comments have been addressed, and in many cases, clarified effectively. I have a few further comments on this revised version.

Author Response and Modification:

Thank you for taking time to review and provide comments on our manuscript once again. We greatly appreciate your additional feedback and have addressed all comments presented. Thanks again.

Reviewer #1: Specific Comments – Overall

1. Use of Theory: Barker/Bronfenbrenner is employed as a form of justification for the study – a purpose I find less convincing unless the authors directly tie back to this framework later in the paper. In the discussion, other, more loosely connected explanatory models are introduced instead. I would still suggest that the findings be discussed in relation to the theories highlighted in the introduction. If not, it may be better to remove these theoretical references from the introduction altogether to avoid confusion.

Author Response and Modification:

1. We appreciate your feedback and have more directly tied the study’s findings back to the ecological theories mentioned in the beginning of the manuscript as follows:

1.1 In the opening paragraph of the Discussion, highlighting the key finding that approximately 70% of children reported participating in an OST organized activity, we suggested this finding may be related to Barker’s Manning Theory and specifically added the following:

Thus, the effects of underpopulation (or “undermanning”) witnessed by Barker and Gump in 1964 may still be relevant today given the high prevalence of participation among rural children and adolescents in these settings [Barker and Gump 1964]. Even if rural children are pressured to participate in OST organized activities out of necessity, participation can significantly impact childhood development, including the improvement of population health PA outcomes [38,71]. (Pages 20-21, lines 410-416)

1.2 We also revised the third paragraph of the Discussion to shift the focus back to the ecological theories mentioned in the beginning of the paper. The revision includes the following:

Our work also revealed no differences in rural children’s organized activity participation classified by family income. Although this finding may not be surprising in communities where “undermanning” exists and settings cannot afford to intentionally (or unintentionally) exclude children from participating, it is noteworthy since youth sport settings have been shown to marginalize children living in rural communities based on social classifications, including those from low-income families [Kellstedt et al. 2021; Von Seggern et al. 2024]. For example, Kellstedt et al. [46] found that FRLS was a significant (p < 0.05) predictor of youth sport participation, in which rural children from full pay families demonstrated almost four times higher participation rates in youth sport compared to those from FRLS families (OR = 3.91, 95% CI = 1.95-7.8) [46]. This suggests that OST non-sport organized activities may be more financially accessible than OST organized youth sport opportunities, particularly for economically disadvantaged children living in rural communities. Cost, however, is not the only factor influencing participation, and within rural communities, non-sport organized activities may be more responsive to different levels of environmental influences on children’s behavior [Bronfenbrenner 1979], such as parental or guardian need for childcare services after the school day (e.g., school-based afterschool programs) if they are commuting out-of-town for employment purposes or their employer does not offer flexible working hour or work-from-home options. Thus, it is necessary to account for different factors influencing OST organized activity participation, including who (e.g., parents, friends, and teachers) and what (e.g., availability, time constraints, and past experiences) are driving children’s participation in these specific behavior settings. This information can uncover why children frequent specific community OST behavior settings within a given social context and could help identify ways to reverse or remove participation barriers and support more equitable organized activity opportunities where all children are encouraged to participate. (Pages 21-22, lines 433-455)

1.3 We also attempted to more explicitly connect social structural differences highlighted in the Discussion to Bronfenbrenner’s ecology theory as follows:

Social structural differences, found within all layers of Bronfenbrenner’s ecological systems theory from the immediate microsystem to the wider macrosystem [Bronfenbrenner 1979], including broader societal gender norms influencing types of opportunities, roles within opportunities, and leader-level practices, may also help to explain sex-based PA disparities [54,88]. (Page 25, lines 512-515)

1.4 In another paragraph in the Discussion, we provided another explanation of one of the study’s findings in relation to Barker and colleagues’ work on behavior settings as follows:

Another possible explanation, consistent with Barker and colleagues’ work on behavior settings, is that in often “undermanned” rural communities, children from lower-income families may feel obligated to participate in OST organized activities out of necessity or a desire for community belonging [Barker and Gump 1964; Barker and Schoggen 1973; Gump and Adelberg, 1978]; in turn, accumulating similar amounts of weekday and weekend OST MVPA with their higher-income participating peers. (Page 26, lines 538-546)

Reviewer #1: Specific Comments – Results/Discussion

2. The presentation of results related to the contribution of OST activities to PA is somewhat confusing. You found no significant differences between participants and non-participants in organised OST activities, yet you suggest (p. 22, line 450) that these activities can still be beneficial for PA. This appears contradictory and should be reviewed.

Author Response and Modification:

2. We have revised the aforementioned paragraph in the Discussion to zoom in on why these activities can still be beneficial for PA, particularly, since we found that children in our study who participated in afterschool programs reported significantly more daily minutes of OST MVPA per weekday than those who did not participate (as presented in the Results section on pg. 16, lines 340-344, and in Fig. 1). As a result, this finding prompted the “unacknowledged shift from OST organized activities to afterschool activities” mentioned in a later comment, but we have attempted to clean this up while making a case for why non-sport OST organized activities can still be beneficial for PA, where we acknowledge any PA is better than no PA (related to Reviewer #2’s first comment). Overall, we combined paragraphs four and five in the Discussion as follows:

Although we found that children who participated in any OST organized activity were not significantly more active during OST on weekdays and weekends than nonparticipants, encouraging participation may still be beneficial given the contribution some OST organized activities have on children’s PA, particularly when participating [81]. For instance, children in the present study who participated in afterschool programs reported significantly more daily minutes of OST MVPA per weekday (meandiff = 3.0 ± 1.2 min/day) than those who did not participate in an afterschool program (p = 0.014). Yet, there was no difference in weekend MVPA between program participants and non-participants, suggesting that afterschool programs may provide a meaningful contribution to PA, but only when children attend. Similarly, a systematic review and meta-analysis by Tassitano et al. [81] found that OST structured settings (i.e., afterschool programs and summer camps) provided youth (aged 3-18 years) with greater amounts of PA during attendance when compared to childcare and school, specifically, 11.7 min/hour and 6.4 min/hour, respectively. Since many (>70%) children in the U.S. are not physically active at the recommended levels [7,8], calls for any increase in childhood PA have been issued, as evidence suggests even small amounts or short episodes of PA provide health benefits both now and in the future (e.g., reduced anxiety symptoms and improved quality of life, respectively) [Piercy et al. 2018; Piercy and Troiano 2018; Sriram et al. 2021]. Therefore, based on the results of our study and others examining the PA patterns of children attending OST structured settings, like afterschool programs, if clubs and other organized activities can schedule time for PA within setting routines, this may positively influence PA promotion efforts. However, scholars warn simply allocating more time in the schedule for PA opportunities may not be sufficient for increasing children’s MVPA. Specifically, the type of PA is also critical (e.g., structured vs. unstructured, individual vs. group) [84], especially in rural communities where evidence indicates non-sport organized activities are less structured and less supervised [25,38]. Thus, inserting shorter bouts of more structured PA oppo

---

## [Decision Letter · Decision Letter 2]

19 Mar 2025

PONE-D-24-23961R2Sociodemographic influences on children’s out-of-school time organized activity participation and physical activity in rural communities: A cross-sectional studyPLOS ONE

Dear Dr. Von Seggern,

Thank you for submitting your manuscript to PLOS ONE. The revised manuscript addressed most of the reviewers' concerns, save for one remaining point related to the conclusion. We invite you to address this remaining point that was raised by both reviewers, and submit a revised version of the manuscript.

We look forward to receiving your revised manuscript.

Kind regards,

Catherine M. Capio

Academic Editor

PLOS ONE

Journal Requirements:

Additional Editor Comments:

Thank you for the revised manuscript, which now addresses nearly all of the comments raised by the reviewers. One last point that needs to be addressed relates to your conclusions, which was noted by both reviewers. If you could please have a look at their comments and improve the conclusion, that would be great.

Reviewers' comments:

Reviewer's Responses to Questions

**Comments to the Author**

1. If the authors have adequately addressed your comments raised in a previous round of review and you feel that this manuscript is now acceptable for publication, you may indicate that here to bypass the “Comments to the Author” section, enter your conflict of interest statement in the “Confidential to Editor” section, and submit your "Accept" recommendation.

Reviewer #1: (No Response)

Reviewer #2: (No Response)

2. Is the manuscript technically sound, and do the data support the conclusions?

Reviewer #1: Yes

Reviewer #2: Yes

3. Has the statistical analysis been performed appropriately and rigorously? 

Reviewer #1: Yes

Reviewer #2: Yes

4. Have the authors made all data underlying the findings in their manuscript fully available?

Reviewer #1: Yes

Reviewer #2: Yes

5. Is the manuscript presented in an intelligible fashion and written in standard English?

Reviewer #1: Yes

Reviewer #2: Yes

6. Review Comments to the Author

Reviewer #1: Thank you for the opportunity to review your manuscript once again. I believe the authors have done a great job addressing and responding to the comments I had on the previous version. It is a thorough piece of work, and they deserve recognition for that.

The only comment I feel has not been fully addressed is comment 2, the one that concerns the conclusions regarding the contribution of after-school activities to MVPA. Since this is cross-sectional data, there is no evidence that after-school activities have contributed to MVPA rather than the other way around. I believe this should at least be mentioned somewhere in the discussion, preferably in the limitations section.

Best of luck with the revision, and I look forward to seeing the paper published.

Reviewer #2: Thank you for revising the manuscript. I believe the conclusion could be more concise to enhance clarity and impact.

7. PLOS authors have the option to publish the peer review history of their article (what does this mean? ). If published, this will include your full peer review and any attached files.

**Do you want your identity to be public for this peer review?** For information about this choice, including consent withdrawal, please see our Privacy Policy .

Reviewer #1: **Yes: ** Johan Högman

Reviewer #2: No

---

## [Author Response · Author response to Decision Letter 3]

25 Apr 2025

Catherine M. Capio, Ph.D.

Academic Editor

PLOS ONE

April 24, 2025

Dear Dr. Capio,

Re: Resubmission of manuscript reference No. PONE-D-24-23961R2

On behalf of my colleagues and myself, thank you for inviting us to address the remaining concern related to the conclusion and submit a revised version of our manuscript titled, “Sociodemographic influences on children’s out-of-school time organized activity participation and physical activity in rural communities: A cross-sectional study,” to PLOS ONE. We also appreciate the time and effort you and each of the reviewers have dedicated to providing additional feedback on ways to improve our manuscript. As a result, we have incorporated changes in the revised manuscript that reflect the comments you and the reviewers have provided in your letter dated March 19, 2025. Similar to our previous revisions, we have attempted to explain changes made in reaction to all comments in a point-by-point fashion shared below, corresponding with the letter’s organization (i.e., Journal Requirements, Additional Editor Comments, and Reviewers’ Comments), and hope that our edits and the responses we have provided below satisfactorily address all the remaining concerns you and the reviewers have noted.

Again, thank you for giving us the opportunity to strengthen our manuscript with your valuable feedback. We look forward to hearing from you regarding our submission and would be glad to respond to any further questions and comments that you and the reviewers may have.

Sincerely,

Mary J. Von Seggern

Part I – Journal Requirements

Comments:

1. Journal Requirements – Reference List

Author Response and Modification:

1. Thank you for bringing this to our attention, and we have reviewed our reference list to ensure it is complete and correct. Additionally, no changes were made to our reference list during this revision process.

Part II – Additional Editor Comments

Editor: General Comments:

Thank you for submitting your manuscript to PLOS ONE. The revised manuscript addressed most of the reviewers' concerns, save for one remaining point related to the conclusion. We invite you to address this remaining point that was raised by both reviewers, and submit a revised version of the manuscript.

Author Response and Modification:

Thank you for taking time to review and provide comments on our revised manuscript. We appreciate your feedback, as well as the opportunity to revise and resubmit the manuscript once again.

Editor: Particular Comments:

1. Thank you for the revised manuscript, which now addresses nearly all of the comments raised by the reviewers. One last point that needs to be addressed relates to your conclusions, which was noted by both reviewers. If you could please have a look at their comments and improve the conclusion, that would be great.

Author Response and Modification:

Thank you for this feedback, and we revised our conclusions based on each reviewer’s feedback as highlighted below:

Reviewer #1 –

1. We addressed Reviewer #1’s comment by acknowledging the limitations of the cross-sectional study design, in which a causal link between participation in after-school activities and MVPA cannot be established. Thus, we added this information to the Limitations section as follows:

Furthermore, we acknowledge the limitation of the cross-sectional study design, in which a causal link between participation in OST non-sport organized activities, including daily afterschool programs, and OST MVPA cannot be established. (Page 28, lines 575-577)

Reviewer #2 –

2. We addressed Reviewer #2’s comment by revising the Conclusions section to enhance clarity and impact as follows:

The present study aimed to examine the sociodemographic factors related to out-of-school time (OST) non-sport organized activity participation and how these factors interact to influence OST physical activity (PA) among children living in rural U.S. communities. The findings suggest that, along with other factors, overall OST non-sport organized activity participation does not play a critical role in maximizing the amount of PA that rural children engage in during OST on weekdays and weekends. Yet, these settings should not be overlooked in PA-promotion efforts provided the fairly high percentage (70%) of rural children participating in non-sport organized activities, coupled with the fact that grade and family income were not a significant driver of participation in this study. Thus, non-sport OST organized activities may be more accessible and responsive to different levels of environmental influences on rural children’s behavior, particularly when considering the well-documented barriers to organized youth sport participation, like cost. More research is needed to explore how these activities could be effectively designed to increase PA since “some physical activity is better than none” [79] and afterschool programs, which often devote time to PA within daily setting routines, can provide a blueprint to consider. Therefore, increasing participation and opportunities for PA within non-sport organized activities could positively impact the MVPA and population health outcomes of children living in rural communities, and advocating for the advancement of an updated National Youth Sports Strategy [42] to include non-sport organized activities may aid in ensuring children nationwide have the opportunity to achieve optimal health. (Pages 28-30, lines 593-622)

3. Additionally, we revised the last sentence in the abstract to improve the conclusion (per previous changes made in the Conclusions section) as follows:

Designing OST organized activity settings to be more accessible and include opportunities for PA may help ensure children can achieve optimal health. (Page 2, lines 61-63)

Part III – Reviewers’ Comments

Reviewer #1: General Comments:

Thank you for the opportunity to review your manuscript once again. I believe the authors have done a great job addressing and responding to the comments I had on the previous version. It is a thorough piece of work, and they deserve recognition for that. Best of luck with the revision, and I look forward to seeing the paper published.

Author Response and Modification:

Thank you for taking time to review our manuscript once again. We greatly appreciate your feedback, well wishes, and additional comments to help strengthen this piece of work. Thanks again.

Reviewer #1: Specific Comments:

1. The only comment I feel has not been fully addressed is comment 2, the one that concerns the conclusions regarding the contribution of after-school activities to MVPA. Since this is cross-sectional data, there is no evidence that after-school activities have contributed to MVPA rather than the other way around. I believe this should at least be mentioned somewhere in the discussion, preferably in the limitations section.

Original Comment #2 - The presentation of results related to the contribution of OST activities to PA is somewhat confusing. You found no significant differences between participants and non-participants in organised OST activities, yet you suggest (p. 22, line 450) that these activities can still be beneficial for PA. This appears contradictory and should be reviewed.

Author Response and Modification:

1. We appreciate your feedback and have more directly addressed this comment by acknowledging the limitations of the cross-sectional study design, in which a causal link between participation in after-school activities and MVPA cannot be established. Thus, we added this information to the Limitations section as follows:

Furthermore, we acknowledge the limitation of the cross-sectional study design, in which a causal link between participation in OST non-sport organized activities, including daily afterschool programs, and OST MVPA cannot be established. (Page 28, lines 575-577)

Reviewer #2: General Comments:

Thank you for revising the manuscript.

Author Response and Modification:

Thank you for taking time to review and provide comments on our manuscript once again. We greatly appreciate your additional feedback to enhance the clarity and impact of our work. Thanks again.

Reviewer #2: Specific Comments:

1. I believe the conclusion could be more concise to enhance clarity and impact.

Author Response and Modification:

1. We appreciate this feedback, as Reviewer #1 and the Editor also mentioned it. Therefore, we have revised our Conclusions section as follows:

The present study aimed to examine the sociodemographic factors related to out-of-school time (OST) non-sport organized activity participation and how these factors interact to influence OST physical activity (PA) among children living in rural U.S. communities. The findings suggest that, along with other factors, overall OST non-sport organized activity participation does not play a critical role in maximizing the amount of PA that rural children engage in during OST on weekdays and weekends. Yet, these settings should not be overlooked in PA-promotion efforts provided the fairly high percentage (70%) of rural children participating in non-sport organized activities, coupled with the fact that grade and family income were not a significant driver of participation in this study. Thus, non-sport OST organized activities may be more accessible and responsive to different levels of environmental influences on rural children’s behavior, particularly when considering the well-documented barriers to organized youth sport participation, like cost. More research is needed to explore how these activities could be effectively designed to increase PA since “some physical activity is better than none” [79] and afterschool programs, which often devote time to PA within daily setting routines, can provide a blueprint to consider. Therefore, increasing participation and opportunities for PA within non-sport organized activities could positively impact the MVPA and population health outcomes of children living in rural communities, and advocating for the advancement of an updated National Youth Sports Strategy [42] to include non-sport organized activities may aid in ensuring children nationwide have the opportunity to achieve optimal health. (Pages 28-30, lines 593-622)

---

## [Editor Report · Decision Letter 3]

27 Apr 2025

Sociodemographic influences on children’s out-of-school time organized activity participation and physical activity in rural communities: A cross-sectional study

PONE-D-24-23961R3

Dear Dr. Von Seggern,

We’re pleased to inform you that your manuscript has been judged scientifically suitable for publication and will be formally accepted for publication once it meets all outstanding technical requirements.

Kind regards,

Catherine M. Capio

Academic Editor

PLOS ONE
---

## [Editor Report · Acceptance letter]

PONE-D-24-23961R3

PLOS ONE

Dear Dr. Von Seggern,

I'm pleased to inform you that your manuscript has been deemed suitable for publication in PLOS ONE. Congratulations! Your manuscript is now being handed over to our production team.

Kind regards,

on behalf of

Dr. Catherine M. Capio

Academic Editor

PLOS ONE